# CERTIFIED WATERMARKS FOR NEURAL NETWORKS

## ABSTRACT

Watermarking is a commonly used strategy to protect creators' rights to digital images, videos and audio. Recently, watermarking methods have been extended to deep learning models – in principle, the watermark should be preserved when an adversary tries to copy the model. However, in practice, watermarks can often be removed by an intelligent adversary. Several papers have proposed watermarking methods that claim to be empirically resistant to different types of removal attacks, but these new techniques often fail in the face of new or better-tuned adversaries. In this paper, we propose the first *certifiable* watermarking method. Using the randomized smoothing technique proposed in Chiang et al., we show that our watermark is guaranteed to be unremovable unless the model parameters are changed by more than a certain $\ell_2$ threshold. In addition to being certifiable, our watermark is also empirically more robust compared to previous watermarking methods.

## 1 INTRODUCTION

With the rise of deep learning, there has been an extraordinary growth in the use of neural networks in various computer vision and natural language understanding tasks. In parallel with this growth in applications, there has been exponential growth in terms of the cost required to develop and train state-of-the-art models (Amodei & Hernandez, 2018). For example, the latest GPT-3 generative language model (Brown et al., 2020) is estimated to cost around $4.6$ million dollars (Li, 2020) in TPU cost alone. This does not include the cost of acquiring and labeling data or paying engineers, which may be even greater. With up-front investment costs growing, if access to models is offered as a service, the incentive is strong for an adversary to try to steal the model, sidestepping the costly training process. Incentives are equally strong for companies to protect such a significant investment.

Watermarking techniques have long been used to protect the copyright of digital multimedia (Hartung & Kutter, 1999). The copyright holder hides some imperceptible information in images, videos, or sound. When they suspect a copyright violation, the source and destination of the multimedia can be identified, enabling appropriate follow-up actions (Hartung & Kutter, 1999). Recently, watermarking has been extended to deter the theft of machine learning models (Uchida et al., 2017; Zhang et al., 2018). The model owner either imprints a predetermined signature into the parameters of the model (Uchida et al., 2017) or trains the model to give predetermined predictions (Zhang et al., 2018) for a certain trigger set (e.g. images superimposed with a predetermined pattern).

A strong watermark must also resist removal by a motivated adversary. Even though the watermarks in (Uchida et al., 2017; Zhang et al., 2018; Adi et al., 2018) initially claimed some resistance to various watermark removal attacks, it was later shown in (Shafieinejad et al., 2019; Aiken et al., 2020) that these watermarks can in fact be removed with more sophisticated methods, using a combination of distillation, parameter regularization, and finetuning. To avoid the cat-and-mouse game of ever-stronger watermark techniques that are only later defeated by new adversaries, we propose the first certifiable watermark: unless the attacker changes the model parameters by more than a certain $\ell_2$ distance, the watermark is guaranteed to remain.

To the best of our knowledge, our proposed watermarking technique is the first to provide a certificate against an $\ell_2$ adversary. Although the bound obtained by the certificate is relatively small, we see it as a first step towards developing watermarks with provable guarantees. Additionally we empirically find that our certified watermark is more resistant to previously proposed watermark removal attacks (Shafieinejad et al., 2019; Aiken et al., 2020) compared to its counterparts – it is thus valuable even when a certificate is not required.

## 2 RELATED WORK

**Watermark techniques** (Uchida et al., 2017) proposed the first method of watermarking neural networks: they embed the watermark into the parameters of the network during training through regularization. However, the proposed approach requires explicit inspection of the parameters for ownership verification. Later, (Zhang et al., 2018; Rouhani et al., 2018) improved upon this approach, such that the watermark can be verified through API-only access to the model. Specifically, they embed the watermark by forcing the network to deliberately misclassify certain "backdoor" images. The ownership can then be verified through the adversary's API by testing its predictions on these images.

In light of later and stronger watermark removal techniques (Aiken et al., 2020; Wang & Kerschbaum, 2019; Shafieinejad et al., 2019), several papers have proposed methods to improve neural network watermarking. (Wang & Kerschbaum, 2019) propose an improved white-box watermark that avoids the detection and removal techniques from (Wang & Kerschbaum, 2019). (Li et al., 2019) propose using values outside of the range of representable images as the trigger set pattern. They show that their watermark is quite resistant to a finetuning attack. However, since their trigger set does not consist of valid images, their method does not allow for black-box ownership verification against a realistic API that only accepts actual images, while our proposed watermark is effective even in the black-box setting.

(Szyller et al., 2019) proposed watermarking methods for models housed behind an API. Unlike our method, their method does not embed a watermark into the model weights itself, and so cannot work in scenarios where the weights of the model may be stolen directly, e.g. when the model is housed on mobile devices.

Finally, (Lukas et al., 2019) propose using a particular type of adversarial example ("conferrable" adversarial examples) to construct the trigger set. This makes the watermark scheme resistant even to the strongest watermark removal attack: ground-up distillation which, starting from a random initialization, trains a new network to imitate the stolen model (Shafieinejad et al., 2019). However, for their approach to be effective, they need to train a large number of models (72) on a large amount of data (e.g. requiring CINIC as opposed to CIFAR-10). While our approach does not achieve this impressive resistance to ground-up distillation, it is also much less costly.

**Watermark removal attacks** However, one concern for all these watermark methods is that a sufficiently motivated adversary may attempt to remove the watermark. Even though (Zhang et al., 2018; Rouhani et al., 2018; Adi et al., 2018; Uchida et al., 2017) all claim that their methods are resistant to watermark removal attacks, such as finetuning, other authors (Aiken et al., 2020; Shafieinejad et al., 2019) later show that by adding regularization, finetuning and pruning, their watermarks can be removed without compromising the prediction accuracy of the stolen model. Wang & Kerschbaum (2019) shows that the watermark signals embedded by (Uchida et al., 2017) can be easily detected and overwritten; (Chen et al., 2019) shows that by leveraging both labeled and unlabeled data, the watermark can be more efficiently removed without compromising the accuracy. Even if the watermark appears empirically resistant to currently known attacks, stronger attacks may eventually come along, prompting better watermark methods, and so on. To avoid this cycle, we propose the first certifiably unremovable watermark: given that parameters are not modified more than a given threshold $\ell_2$ distance, the watermark will be preserved.

**Certified defenses for adversarial robustness** Our work is inspired by recent work on certified adversarial robustness, (Cohen et al., 2019; Chiang et al., 2019; Wong & Kolter, 2017; Mirman et al., 2018; Weng et al., 2018; Zhang et al., 2019; Eykholt et al., 2017; Levine & Feizi, 2019). Certified adversarial robustness involves not only training the model to be robust to adversarial attacks under particular threat models, but also proving that no possible attacks under a particular constraint could possibly succeed. Specifically, in this paper, we used the randomized smoothing technique first developed by (Cohen et al., 2019; Lecuyer et al., 2019) for classifiers, and later extended by (Chiang et al., 2020) to deal with regression models. However, as opposed to defending against an $\ell_2$-bounded threat models in the image space, we are now defending against an $\ell_2$-bounded adversary in the parameter space. Surprisingly, even though the certificate holds only when randomized smoothing is applied, empirically, when our watermark is evaluated in a black-box setting on the non-smoothed model, it also exhibits stronger persistence compared to previous methods.

## 3 METHODS

Below, we introduce the formal model for neural network watermarking, and the watermark removal adversaries that we are concerned with. Then, we describe some background related to randomized smoothing, and show that by using randomized smoothing we can create a watermark that provably cannot be removed by an $\ell_2$ adversary.

### 3.1 WATERMARKING

**White box vs black box**  We first introduce the distinction between black box and white box settings from the perspective of the owner of the model. In a white box setting, parameters are known. In a black box setting, the model parameters are hidden behind an API. We consider cases where the owner may have either black box or white box access to verify their watermarks.

**Black-box watermarking**  In backdoor-based watermarking, the owner employs a "trigger set" of specially chosen images that has disjoint distribution compared to the original dataset. If another model makes correct predictions on this trigger set, then this is evidence that the model has been stolen. A backdoor-based watermark can be verified in a black-box setting.

The trigger set may be chosen in various ways. (Zhang et al., 2018) considered three different methods of generating the trigger set: embedded content, pre-specified noise, and abstract images. Embedded content methods embed text over existing datasets and assigns all examples with the text overlay the same fixed label. Pre-specified noise overlays Gaussian noise on top of existing dataset and again assigns the examples with the same fixed label. For abstract images, a set of images from a different domain is additionally used to train the network. For example, MNIST images could form the trigger set for a CIFAR-10 network, so if an adversary's model performs exceedingly well on MNIST images, then the adversary must have used the stolen model. Examples of trigger set images are presented in Figure 1.

Our proposed method builds upon such backdoor-based watermarks, so our marked model can also naturally be verified in the black-box manner even though our certificate is only valid in the white-box setting described in the next section.

**White-box watermarking**  White-box watermarks in general embed information directly into the parameters. Our proposed watermark does not directly embed information into parameters, but parameter access is required for verification, which makes our proposed method a *white-box* watermark. The rationale for using such a white-box watermark is detailed below.

In the black-box setting, to verify model ownership, we generally check that the trigger set accuracy function from parameters to accuracy $f(\theta)$ is larger than a threshold (Shafieinejad et al., 2019). The trigger set accuracy function takes in model parameter as input and outputs the accuracy on the trigger set. Since directly certifying the function is hard, we first convert the trigger set accuracy function $f(\theta)$ to its smoothed counterpart $h(\theta)$, and then check that $h(\theta)$ is greater than the threshold $t$ for ownership verification. Practically, one converts the base function to the smoothed function by injecting random noise into the parameters during multiple trigger set evaluations, and then taking the median trigger set accuracy as $\hat{h}$. Note that this verification process *requires* access to parameters, so ownership verification using $\hat{h}$ is considered a *white-box* watermark.

**Watermark Removal Threat Model**  In our experiments, we consider three different threat models to the watermark verification: 1) distillation, 2) finetuning, and 3) an $\ell_2$ adversary.

In the distillation threat model (1), we assume that the adversary initializes their model with our original model, and then trains their model with distillation using unlabeled data that comes from the same distribution. In other words, the adversary uses our original model to label the unlabeled data for finetuning. (Shafieinejad et al., 2019) propose first adding some regularization during the initial part of the attack to remove the watermark, and then later turning off the regularization to fully recover the test accuracy of the model. We experiment with this distillation attack both with and without regularization.

In the finetuning threat model (2), the adversary has its own labeled dataset from the original data-generating distribution. This adversary is strictly stronger compared to the distillation threat model. In our experiments, we make the conservative assumption that the adversary has exactly the same amount of data as the model owner.

The $\ell_2$ adversary (3) obtains the original model parameters, and then is allowed to move the parameters at most a certain $\ell_2$ distance to maximally decrease trigger set accuracy. Even though the $\ell_2$ adversary is not a completely realistic threat model, we argue similarly to the adversarial robustness literature (Carlini et al., 2019) that being able to defend against a small $\ell_2$ adversary is a requirement for defending against more sophisticated attacks. In our experiments, we empirically find that a large shift of parameters in $\ell_2$ distance is indicative of the strength of the adversary. For example, training the models for more time, with a larger learning rate, or using ground truth labels as opposed to distillation are all stronger attacks, and as expected, they both remove the watermark faster and move the parameters by a greater $\ell_2$ distance (Table 2). Additionally, given a local Lipschitz constant of $L$ and a learning rate of $r$, the number of steps required to move outside of the $\epsilon$-$\ell_2$ ball can be upper bounded by $\epsilon/(rL)$, and we think the number of steps is a good proxy to the computational budget of the adversary.

## 3.2 WATERMARK CERTIFICATION

For our certificates, we focus on the $\ell_2$ adversary described above: the goal of certification is to bound the worst-case decrease in trigger set accuracy, given that the model parameters do not move too far in $\ell_2$ distance. Doing this directly is in general quite difficult (Katz et al., 2019), but using techniques from (Chiang et al., 2020; Cohen et al., 2019), we show that by adding random noise to the parameters it is possible to define a smoothed version of the model and bound the change in its trigger set accuracy.

**Deriving the certificate**   Before we start describing the watermark certificate, we will first introduce the percentile smoothed function from (Chiang et al., 2020).

**Definition 1** *Given $f : \mathbb{R}^d \rightarrow \mathbb{R}$ and $G \sim N(0, \sigma^2 I)$, we define the* percentile smoothing *of $f$ as*

$$\underline{h}_p(x) = \sup\{y \in \mathbb{R} \mid \mathbb{P}[f(x + G) \leq y] \leq p\} \tag{1}$$

$$\overline{h}_p(x) = \inf\{y \in \mathbb{R} \mid \mathbb{P}[f(x + G) \leq y] \geq p\} \tag{2}$$

As mentioned in (Chiang et al., 2020), the two forms $\underline{h}_p$ and $\overline{h}_p$ are needed to handle edge cases with discrete distributions. While $h_p$ may not admit a closed form, we can approximate it by Monte Carlo sampling (Cohen et al., 2019).

There are some differences from existing adversarial robustness work in how we apply these bounds. First, while the robustness literature applies the smoothing results to bound outputs of the classifier itself, we apply smoothing over the trigger set accuracy function to bound changes in trigger set accuracy. Second, we are applying smoothing over parameters as opposed to input. Our trigger set accuracy function $f(X, \theta)$ in general takes in two arguments: $X$, a set of images, and $\theta$, the model parameters. In the case of adversarial robustness, the model parameters $\theta$ are constant after training while the attacker perturbs the image $x$. But in our case, the trigger set $X$ remains constant and the adversary can only change $\theta$. Therefore, to defend against our specific adversary, we apply smoothing over $\theta$ as opposed to $X$. Since the trigger set $X$ is constant for our case, we simply write the trigger set accuracy function as $f(\theta)$ for the remaining part of the paper.

In our proposed watermark, we use the median smoothed version ($h_{50\%}$) of the trigger set accuracy function for ownership verification. Empirically evaluating $h_{50\%}$ essentially involves adding noise to several copies of the model parameters, calculating trigger set accuracy for all of them, and taking the median trigger set accuracy. The details of evaluating smoothed trigger set accuracy are described in Algorithm 1

Even though the evaluation process of $h_{50\%}$ is more involved compared to the base trigger set accuracy function, the smoothed version allows us to use Lemma 1 from (Chiang et al., 2020) to bound the worst case change in the trigger set accuracy given bounded change in parameters, as shown in Corollary 1.

**Corollary 1** *A median-smoothed function $h_{50\%}$ with adversarial perturbation $\delta$ can be bounded as*

$$\underline{h}_{\Phi(-\epsilon/\sigma)}(\theta) \leq h_{50\%}(\theta + \delta) \quad \forall \, \|\delta\|_2 < \epsilon, \tag{3}$$

*where $\Phi$ is the standard Gaussian CDF.*

Using the above corollary, we can then bound the worst case trigger set accuracy given the $\epsilon$ adversary by evaluating $\underline{h}_{\Phi(-\epsilon/\sigma)}(x)$. Even though $\underline{h}_{\Phi(-\epsilon/\sigma)}(x)$ does not have a closed form, we can calculate an empirical estimator that would lower bound it with sufficient confidence $c$. We detail steps for calculating the estimator in Algorithm 1.

---

**Algorithm 1** Evaluate and Certify the Median Smoothed Model

---

  **function** TRIGGERSETACCURACY($f, \theta, \sigma, n$)
    $\hat{w} \leftarrow$ AddGaussianNoise($\theta, \sigma, n$)            ▷ $n$ simulations of noised parameter $w$
    $\hat{a} \leftarrow f(\hat{\theta})$            ▷ evaluate trigger accuracy for each simulation of $w$
    $\hat{a} \leftarrow$ Sort($\hat{a}$)            ▷ Sort simulated accuracies
    $a_{median} \leftarrow \hat{a}_{\lfloor 0.5n \rfloor}$            ▷ Take the median
    **return** $a_{median}$
  **function** TRIGGERSETACCURACYLOWERBOUND($f, \theta, \sigma, \epsilon, n, c$)
    $\hat{\theta} \leftarrow$ AddGaussianNoise($\theta, \sigma, n$)            ▷ $n$ simulations of noised parameter $w$
    $\hat{a} \leftarrow f(\hat{\theta})$            ▷ evaluate trigger accuracy for each simulation of $\theta$
    $\hat{a} \leftarrow$ Sort($\hat{a}$)            ▷ Sort simulated accuracies
    $k \leftarrow$ EmpiricalPercentile($n, c, \sigma, \epsilon$)            ▷ Algorithm 3 in Appendix
    $\underline{a} \leftarrow \hat{a}_k$            ▷ $\hat{a}_k$ Lower bound $\underline{h}_{\Phi(-\epsilon/\sigma)}(\theta)$ with confidence $c$
    **return** $\underline{a}$

---

**Algorithm 2** Embed Certifiable Watermark

---

  **Required:** training samples $X$, trigger set samples $X_{trigger}$, learning rate $\tau$, maximum noise level $\epsilon$, replay count $k$, noise sample count $t$
  **for** epoch = 1, ... , N **do**
    **for** $B \subset X$ **do**
      $g_\theta \leftarrow E_{(x,y) \in B}[\nabla_\theta l(x, y, \theta)]$
      $\theta \leftarrow \theta - \tau g_\theta$
    **for** $B \subset X_{trigger}$ **do**
      $g_\theta = 0$
      **for** $i = 1$ to $k$ **do**
        $\sigma \leftarrow \frac{i}{k}\epsilon$
        **for** $j = 1$ to $t$ **do**
          $G \sim N(0, \sigma^2 I)$
          $g_\theta \leftarrow g_\theta + E_{(x,y) \in B}[\nabla_\theta l(x, y, \theta + G)]$
      $g_\theta \leftarrow g_\theta / (kt)$
      $\theta \leftarrow \theta - \tau g_\theta$

---

**Embedding the Certifiable Watermark** To embed the watermark during training, we add Gaussian noise and train on the trigger set images with the desired labels. For a given trigger set image, we average gradients across several (in our experiments, 100) draws of noise to better approximate the gradient of the smoothed classifier. Directly adding a large amount of noise into all parameters makes training unstable, so we incrementally increase the levels of noise within each epoch. In our experiments, we inject Gaussian noise with a range of standard deviations $\sigma$ ranging from 0 to 1. Empirically, we notice that the test accuracy drops when using this technique to embed the watermark, so to recover some of the lost test accuracy, we warm up the model with regular training and only begin embedding the watermark after the fifth epoch. We note that using warm-up epochs to recover clean accuracy is a common practice in the robustness literature (Balaji et al., 2019; Gowal et al., 2018). The detailed training method is described in Algorithm 2.

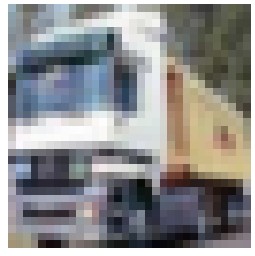 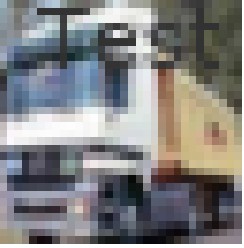 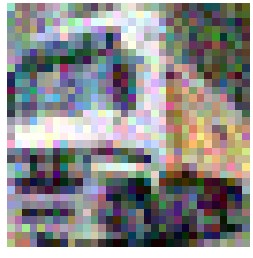 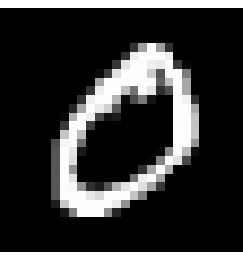

(a) Original      (b) Embedded Content      (c) Gaussian Noise      (d) Unrelated

Figure 1: Samples of the backdoor images used for watermarking.

## 4 EXPERIMENTS

In our first set of experiments, we investigate the strength of our certificate under two datasets and three watermark schemes. In our second set of experiments, we evaluate the watermark's empirical robustness to removal compared to previous methods that claimed resistance to removal attacks.

### 4.1 EXPERIMENTAL SETTINGS

To produce the trigger sets themselves, we consider the three schemes from Zhang et al. (2018): images with embedded content (superimposed text), images with random noise, or images from an unrelated dataset (CIFAR-10 for MNIST and vice versa) (Figure 1). While we generated certificates for all three schemes, we focus on embedded content watermark for empirical persistency evaluation.

To train the watermarked model, we used ResNet-18, SGD with learning rate of .05, momentum of .9, and weight decay of 1e-4. The model is trained for 100 epochs, and the learning rate is divided by 10 every 30 epochs. Only 50% of the data is used for training, since we reserve the other half for the adversary. For our watermark models, we select $\sigma$ of 1, replay count of 20, and noise sample count of 100. For certification, we use 10000 instances of Monte Carlo sampling to perform smoothing.

To attack the model, we used Adam with learn rates of .1, .001 or .0001 for 50 epochs. We tested three different types of attacks: finetuning, hard-label distillation, and soft-label distillation. Soft-label distillation takes the probability distribution of the original model as labels, whereas hard-label distillation takes only the label with maximum probability. We always give the adversary the same amount of data as the owner (labeled for finetuning, unlabeled for distillation) to err on the conservative side for our evaluation.

### 4.2 WATERMARK CERTIFICATE

In this section, we investigate the certified trigger set accuracy that our watermarking is able to guarantee against $\ell_2$ adversaries of various strengths. To contextualize the meaning of a certified $\ell_2$ radius, we consider the size of the empirical changes in parameters observed after performing various watermark removal attacks.

As shown in Table 1, we are able to obtain nontrivial trigger set accuracy certificate for radius up to 0.4 for all datasets and watermark schemes considered. Our certificate seems to be similarly effective across all trigger set types. In the best scenario for CIFAR-10, we can certify that the trigger set accuracy does not drop below 51% as long as parameters do not move more than an $\ell_2$ distance of 1.

To see how long our certificates can persist in the face of attack, we measure the approximate amount of $\ell_2$ parameter change in the first epoch under different attack settings. In Table 2, with learning rate 0.0001, parameters change by $\ell_2$ distance of approximately 2-3. In other words, it would require approximately 1/3 to 1/2 of an epoch to move outside of a certified radius of 1. (We focus here on the first epoch because changes are relatively small in succeeding epochs; see appendix.)

Interestingly, attacks considered to be stronger correspond to changes of a greater distance. This relationship helps support the use of $\ell_2$ radius as a proxy for the strength of the adversary. For example, fine-tuning has been found to be a stronger attack compared to hard label distillation, and

| | | $\ell_2$ Radius ($\epsilon$) | | | | | |
|---|---|---|---|---|---|---|---|
| Dataset | Watermark | 0.2 | 0.4 | 0.6 | 0.8 | 1 | 1.2 |
| MNIST | Embedded content | 100% | 95% | 47% | 3% | 0% | 0% |
| MNIST | Noise | 100% | 91% | 7% | 0% | 0% | 0% |
| MNIST | Unrelated | 100% | 94% | 45% | 4% | 0% | 0% |
| CIFAR-10 | Embedded content | 100% | 100% | 100% | 93% | 51% | 5% |
| CIFAR-10 | Noise | 100% | 100% | 100% | 100% | 47% | 0% |
| CIFAR-10 | Unrelated | 100% | 100% | 100% | 97% | 35% | 0% |

Table 1: Certified trigger set accuracy at different radius

| Attack Type | Finetuning | Distillation Hard Label | Distillation Soft Label | Finetuning | Distillation Hard Label | Distillation Soft Label |
|---|---|---|---|---|---|---|
| Learning Rate | 0.0001 | 0.0001 | 0.0001 | 0.001 | 0.001 | 0.001 |
| MNIST | 2.67 | 2.39 | 1.56 | 19.39 | 17.58 | 20.35 |
| CIFAR-10 | 2.85 | 2.41 | 2.06 | 19.93 | 19.40 | 19.29 |

Table 2: $\ell_2$ distance change in the first epoch

correspondingly (Shafieinejad et al., 2019), fine-tuning moves the network by a larger distance in the first epoch compared to hard label distillation. Similarly, an attack that is stronger due to a higher learning rate moves the parameters much faster compared to an attack with a lower learning rate.

Overall, it would take approximately 0.03 to 0.3 epochs for the attacker to escape the certified radius, depending on the type of attack, watermark schemes, and dataset. Our certified bounds are not trivial, but they are still quite small compared to what would be realistically useful – a common problem involving certified properties of neural networks which can hopefully be remedied with improved training and improved certification techniques. In the next section, we show that even though our certificates are quite small, the watermarks are empirically stronger than the certificate is able to guarantee: in most cases, our watermarks are more resistant to removal attacks compared to previous methods in both the white-box and black-box settings.

### 4.3 EMPIRICAL WATERMARK PERSISTENCE EVALUATION

In this section, we evaluate the persistence of our proposed watermarking methods and the model's performance on the original dataset. For all experiments in this section, we use the embedded content method to produce the trigger set. We compare our watermark method with the baseline method from Zhang et al. (2018), which is the same as our watermark method but without noise injection during training.

For persistence evaluation, we focus on two main attacks: the distillation attack and the finetuning attack, as both of these have been shown to be very effective in (Shafieinejad et al., 2019; Aiken et al., 2020). In addition, we tested the effect of different learning rates and label smoothing levels, which have also been shown to influence the effectiveness of watermark removal techniques Shafieinejad et al. (2019). To make our attacks more similar to Shafieinejad et al. (2019), we also experimented with adding parameter regularization during attack.

We first evaluate our proposed watermark against finetuning attacks. In Table 3, we see that our proposed watermark is much more robust with respect to finetuning attacks than the baseline method on CIFAR-10, and is comparably resistant on MNIST. In the case of CIFAR-10, the baseline watermark is completely removed within less than 10 epochs (See Figure 2 in Appendix), but our white-box watermark is still visible after finetuning for up to 50 epochs. In the case of MNIST, both the proposed method and the baseline are quite resistant. However, our proposed method achieves slightly higher trigger set accuracy for both white-box watermarks and black-box watermarks throughout the 50 epochs of the finetuning attack.

In the face of the distillation attack, we find our white-box watermark to be extremely resistant. The trigger set accuracy remains 100% even after 50 epochs of attack. However, our black-box watermark works more effectively on CIFAR-10 than MNIST. In the case of CIFAR-10, the black-

| Dataset | Attack | lr | Baseline Watermark | Black-box Watermark | White-box Watermark |
|---------|--------|-----|-----------|-----------|-----------|
| MNIST | Finetuning | 0.0001 | 45.31% | 59.38% | 100.00% |
| MNIST | Finetuning | 0.001 | 50.00% | 54.70% | 100.00% |
| MNIST | Hard-Label Distillation | 0.001 | 42.19% | 50.00% | 100.00% |
| MNIST | Soft-Label Distillation | 0.001 | 96.88% | 100.00% | 100.00% |
| CIFAR-10 | Finetuning | 0.0001 | 17.20% | 9.40% | 100.00% |
| CIFAR-10 | Finetuning | 0.001 | 14.06% | 10.94% | 100.00% |
| CIFAR-10 | Hard-Label Distillation | 0.001 | 29.69% | 81.25% | 100.00% |
| CIFAR-10 | Soft-Label Distillation | 0.001 | 81.25% | 100.00% | 100.00% |
| MNIST | Hard-Label Distillation + Reg | 0.1 | 40.63% | 32.81% | 0.00% |
| CIFAR-10 | Hard-Label Distillation + Reg | 0.1 | 8.00% | 27.00% | 0.00% |

Table 3: Trigger set accuracy after 50 epochs of removal attacks. We note that this is only a snapshot of the trigger set accuracy. During training, trigger set accuracies could sometimes fluctuate significantly (see figures in Appendix). We use watermarks from Zhang et al. (2018) as the baseline watermark.

box watermark remains at 81.25% after 50 epochs of distillation attack, whereas only 50.00% of trigger set accuracy remains for MNIST.

When regularization is added in addition to distillation, we find that our white-box watermark is completely removed. This could be due to regularization moving the parameters further in terms of $\ell_2$ norm. However, we note that our black-box watermark still persists similarly to the baseline.

In some cases, the baseline watermark persists quite strongly. For example, in the case of soft-label distillation, the baseline watermark still achieves higher than 75% accuracy after attack. We tried a variety of settings, but we had difficulty completely removing the watermark as described in (Shafieinejad et al., 2019). Differences in performance could be due to architecture, regularization, or other factors – experimental code was not released by (Shafieinejad et al., 2019), so it is hard to know exactly what might be the cause. However, we note that our main goal is to show that our proposed watermark is more resistant to removal, and our trigger set accuracy is consistently higher compared to the baseline throughout the attack.

Even though our watermark is generally more resistant in both the white-box and black-box settings, our proposed method does slightly decrease the accuracy of the model on the original dataset. Test accuracies are decreased by 0.1% (from 99.5% to 99.4%) and 3.3% (from 89.3% to 86.0%) for MNIST and CIFAR-10 respectively. We note that the decrease in clean accuracy has been historically observed for other forms of robust training (Madry et al., 2017), and recovery of the test accuracy in robust training is still an active area of research (Balaji et al., 2019).

## 5 CONCLUSION

We present the first (to our knowledge) certifiable neural network watermark – trigger set accuracy is provably maintained unless the network parameters are moved by more than a given $\ell_2$ distance. The certificates are in practice somewhat small, and the threat model considered is somewhat narrow, but we see this as the first step towards guaranteed persistence of watermarks in the face of adversaries – a valuable property in real-world applications. Future improvements in theory could result in tighter bounds, and future improvements in training and architecture could result in better certificates from the existing bounds.

At the same time, we find that our certifiable watermarks are empirically far more resistant to removal than the certified bounds can guarantee. Indeed in the face of the removal attacks from the literature, our watermarks are more persistent than previous methods. Our randomized-smoothing-based training scheme is therefore a watermarking technique of interest even where a certificate is not needed. We are hopeful that our technique represents a contribution to both the theory and practice of neural network watermarking, and that this approach can lead to watermarks that are both empirically useful while coming with provable guarantees.

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

# A APPENDIX - TRIGGER SET TRAJECTORIES DURING ATTACK

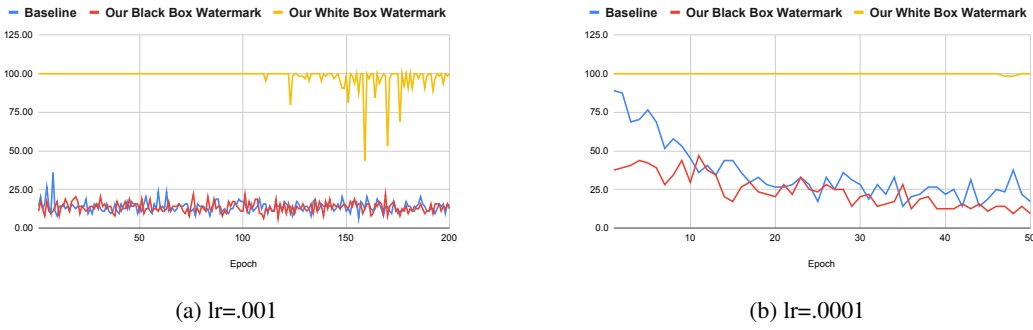

(a) lr=.001

(b) lr=.0001

Figure 2: CIFAR-10 trigger set accuracy when faced with finetuning attacks

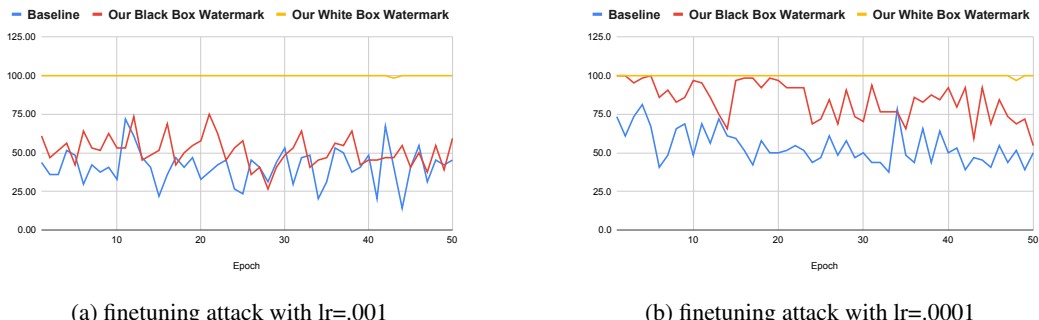

(a) finetuning attack with lr=.001

(b) finetuning attack with lr=.0001

Figure 3: MNIST trigger set accuracy when faced with finetuning attacks

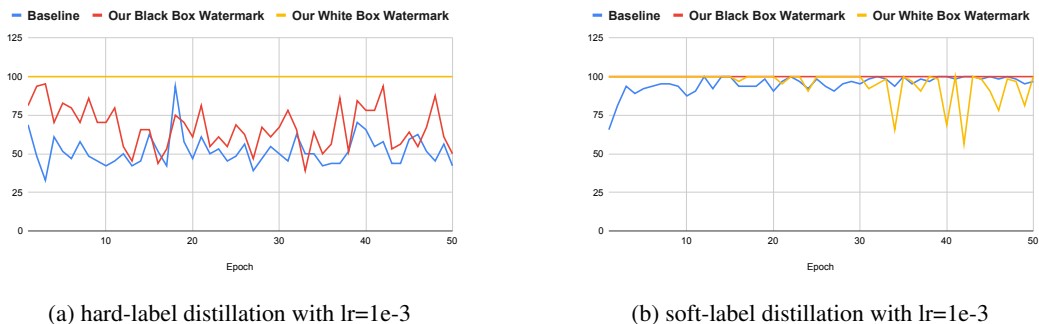

(a) hard-label distillation with lr=1e-3

(b) soft-label distillation with lr=1e-3

Figure 4: MNIST trigger set accuracy when faced with distillation attacks

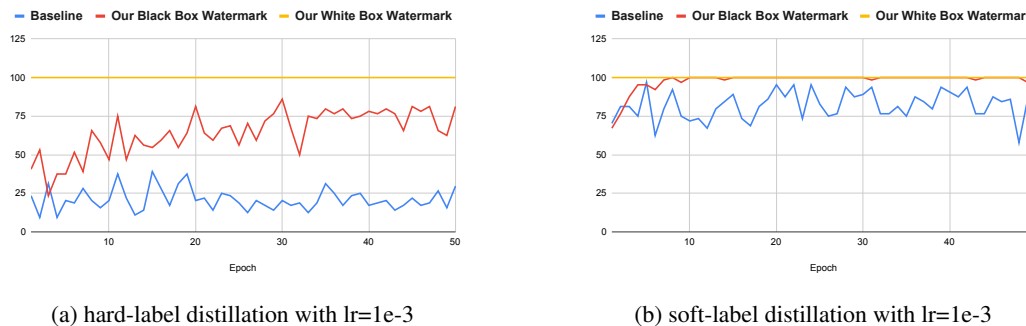

(a) hard-label distillation with lr=1e-3        (b) soft-label distillation with lr=1e-3

Figure 5: CIFAR-10 trigger set accuracy when faced with distillation attacks

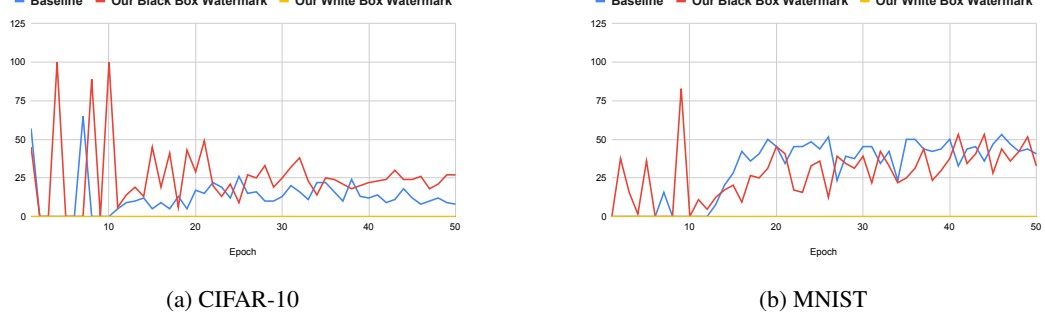

(a) CIFAR-10             (b) MNIST

Figure 6: Trigger set accuracy when faced with distillation+regularization attacks

# B APPENDIX - ALGORITHM FOR EMPIRICAL ORDER STATISTIC

---

**Algorithm 3** Choosing the empirical order statistics that sufficiently lower bound the theoretical percentile

---

**function** EMPIRICALPERCENTILE($n, c, \sigma, \epsilon$)

    $p_{lower} \leftarrow \Phi(-\frac{\epsilon}{\sigma})$         ▷ calculate theoretical percentile that we should be lower bounding

    $\underline{\hat{K}}_{lower}, \overline{\hat{K}}_{lower} \leftarrow 0, \lfloor n \cdot p_{lower} \rfloor$     ▷ initialized empirical order statistics for lower bound

    **while** $\overline{\hat{K}}_{lower} - \underline{\hat{K}}_{lower} > 1$ **do**

        $\dot{K}_{lower} \leftarrow \lfloor (\overline{\hat{K}}_{lower} + \underline{\hat{K}}_{lower})/2 \rfloor$

        **if** 1-Binomial(n, $\dot{K}_{lower}$, $p_{lower}$) > c **then**

            $\underline{\hat{K}}_{lower} \leftarrow \dot{K}_{lower}$

        **else**

            $\overline{\hat{K}}_{lower} \leftarrow \dot{K}_{lower}$

    **if** $\underline{\hat{K}}_{lower} > 0$ **then**

        **return** $\underline{\hat{K}}_{lower}$

    **else**

        **return** null

---

## C    APPENDIX - $\ell_2$ NORM CHANGE DURING ATTACK

| Method | 1st | 2 | 3 | 4 | 5 | 6 | 7 | 8 | 9 | 10 |
|---|---|---|---|---|---|---|---|---|---|---|
| **CIFAR** | | | | | | | | | | |
| Hard label $10^{-4}$ | 2.41 | 3.07 | 3.56 | 4.00 | 4.37 | 4.71 | 5.00 | 5.32 | 5.64 | 5.88 |
| Hard label $10^{-3}$ | 19.4 | 21.33 | 23.45 | 25.71 | 27.95 | 30.02 | 32.06 | 34.06 | 36.12 | 38.04 |
| Soft label $10^{-4}$ | 2.06 | 2.47 | 2.73 | 2.95 | 3.2 | 3.47 | 3.73 | 3.97 | 4.16 | 4.38 |
| Soft label $10^{-3}$ | 19.29 | 20.19 | 21.00 | 21.9 | 22.75 | 23.7 | 24.64 | 25.5 | 26.36 | 27.34 |
| Finetune $10^{-4}$ | 2.85 | 3.47 | 4.18 | 4.79 | 5.48 | 6.13 | 6.76 | 7.37 | 7.92 | 8.45 |
| Finetune $10^{-3}$ | 19.93 | 22.57 | 25.54 | 28.41 | 31.34 | 34.31 | 37.31 | 40.18 | 42.98 | 45.73 |
| **MNIST** | | | | | | | | | | |
| Hard label $10^{-4}$ | 2.39 | 3.14 | 3.71 | 4.17 | 4.66 | 5.04 | 5.32 | 5.63 | 5.92 | 6.25 |
| Hard label $10^{-3}$ | 17.58 | 19.34 | 21.2 | 22.87 | 24.77 | 26.73 | 28.77 | 30.33 | 32.12 | 33.83 |
| Soft label $10^{-4}$ | 1.56 | 2.23 | 2.86 | 3.46 | 3.98 | 4.45 | 4.94 | 5.35 | 5.76 | 6.15 |
| Soft label $10^{-3}$ | 20.35 | 22.51 | 25.00 | 28.12 | 30.29 | 32.31 | 34.35 | 36.58 | 38.84 | 41.1 |
| Finetune $10^{-4}$ | 2.67 | 3.44 | 4.08 | 4.61 | 5.12 | 5.67 | 6.03 | 6.45 | 6.87 | 7.22 |
| Finetune $10^{-3}$ | 19.4 | 21.33 | 23.43 | 25.53 | 27.59 | 29.78 | 31.96 | 34.15 | 36.33 | 38.1 |

Table 4: Difference in $\ell_2$ norm from previous parameters after each epoch of attack. After the first epoch, the increase is general small on each successive epoch.

