# OpenReview forum: "Certified Watermarks for Neural Networks"
_ICLR.cc/2021/Conference — Reject_

### Official Review · AnonReviewer1 · 2020-10-26
**This paper present the first certifiable neural network watermark method.**

**Rating:** 5
**Confidence:** 4

**Review:**

Comments:
This paper present the first certifiable neural network watermark method. By extending method proposed by Chiang et.al. [1] to the watermark embedding and extraction process, it is possible to ensure that the watermark is robust to watermark removal when the network parameters are modified by less than a certain calculated value. Specifically, the proposed method adds Gaussian noises to parameters instead of images. Overall, the idea of making a provable model watermark is novel.

Advantages:
1. The article is well-written and gives a detailed description of the related work, as well as a clear flowchart of the algorithm. 2. The concept of certifying neural network parameters is novel, and it works under a lot of model parameter modification based attacks.  Normally as a defenser, we will train a robust classifier to defend against adversarial example, and the perturbation happens on the testing images. In this case, the perturbation happens on the pretrained released neural networks.

Concerns:
1. The technical contribution of this paper is a bit weak in that they mostly followed [1] and the only interesting point is the expansion into the new scene of model watermarking.
2. I think there is a lack of experiments on the robustness and model performance of watermarks under different $\epsilon$.
3. CIFAR-10 and MNIST are both small datasets. The authors should provide results on larger benchmark dataset, e.g. ImageNet for verifying the method, as well as providing comparison results between a large dataset and a small dataset.

[1] Chiang, P., Curry, M.J., Abdelkader, A., Kumar, A., Dickerson, J.P., & Goldstein, T. (2020). Detection as Regression: Certified Object Detection by Median Smoothing. ArXiv, abs/2007.03730.

---

> ### Author Response · Authors · 2020-11-21
> **Review Response**
>
> **“The technical contribution of this paper is a bit weak in that they mostly followed [1] and the only interesting point is the expansion into the new scene of model watermarking.”**
>
> We would like to emphasize that making this approach work requires more than just borrowing the idea of Gaussian smoothing. Directly training with large noise in the parameter space is very unstable, so we propose an epsilon training schedule to make certificate training more effective. To retain test accuracy, we propose adding noise to the parameters only for the watermark examples and not regular examples. This is distinct from the prior Gaussian smoothing paper, which applies smoothing to all images. It is only with the combination of techniques that our proposed defense empirically outperforms other (non-certified) schemes.
>
> Finally, we would like to emphasize the main contribution of the paper is the conceptual framework that it is possible to certify a watermark for neural networks. Similar to the verification and robustness literature, earlier contributions often use simpler techniques, but over time, more sophisticated methods are developed to tailor results to the particular application.
>
> **“I think there is a lack of experiments on the robustness and model performance of watermarks under different  ϵ.”**
>
> We did provide certified trigger set accuracy for different ϵ in Table 1. Perhaps this refers to robustness and model performance of watermarks under different noise levels? We ran additional experiments that analyze the performance of the model as we change the level of the noise below for CIFAR-10. According to our experiments, as one makes the model more robust against watermark removal,  the model’s performance decreases. This trade-off is similar to the trade-off observed in adversarial robustness literature. As the level of noise increases, training also becomes more unstable. For example, using the same hyperparameters as our other experiments in the paper, we were unable to train models with Sigma = 1.5.
>
> --------
> Table 1 - Trade-off between test accuracy and certified watermark accuracy
>
> Sigma/Certified Watermark Accuracy (Radius 0.2-0.4-0.6-0.8-1-1.2-1.4)
>
> 1/100-100-100-93-51-5-0
>
> 1.1/100-100-100-97-63-13-0
>
> 1.2/100-100-100-100-98-74-24
>
> Sigma/Test Accuracy
>
> 1/**86.00**
>
> 1.1/84.56
>
> 1.2/84.18
>
> ---------
>
> However, this is not to say that it is impossible to train a model with Sigma = 1.5. We did find an alternative setting where Sigma = 1.5 is trainable and offers higher robustness compared to Sigma = 1.0-1.2. However, since the hyperparameters are not the same, we do not list the results here as we don’t think they are directly comparable.
>
> **“CIFAR-10 and MNIST are both small datasets. The authors should provide results on larger benchmark dataset, e.g. ImageNet for verifying the method, as well as providing comparison results between a large dataset and a small dataset.”**
>
> Due to time and computational constraints, we were only able to run results for CIFAR-100.
>
> Again, we similarly observe a drop in accuracy from 68.28% to 67.23%. However, we do note that the drop in accuracy is comparable to the drop in accuracy for CIFAR-10, and does not deteriorate more severely when moving on to a more challenging dataset.
> In the blackbox setting, our method outperforms the baseline method slightly most of the time. However, in the attacks that we tested below, our whitebox watermark significantly outperforms the previous method.
>
> ----------
> Table 2 - Watermark retention on CIFAR-100
>
> CIFAR 100 - text_overlay
>
> Attack/LR/Baseline Blackbox Watermark/Our Blackbox Watermark/Our Whitebox Watermark
>
> Hard - Label Distillation/0.001/7.8125/12.5/50
>
> Hard - Label Distillation/0.01/3.125/1.5625/100
>
> Soft - Label Distillation/0.001/96.875/96.875/98.4375
>
> Soft - Label Distillation/0.01/1.56/12.5/95.3125
>
> Fine-Tune Distillation/0.0001/18.75/23.4375/100
>
> ----------

---

> > ### Comment · AnonReviewer1 · 2020-11-24
> > **Response**
> >
> > Thank you for your additional explanations as well as response to my questions. I still feel like the contribution is somewhat minor and the proposed algorithm is not practical as expected. Besides, we miss the results in terms of datasets with larger image size, e.g. mini-ImageNet or ImageNette. Thanks again for your hard work, but I will keep my score unchanged.

---

### Official Review · AnonReviewer4 · 2020-10-28
**Recommend to reject**

**Rating:** 4
**Confidence:** 4

**Review:**

The authors have created a well written paper for a new watermarking method that addresses an important challenge in security intellectual property rights for deep learning models. They claim their method has resistance to l2 attacks within a certifiable bound, and show experimental results that the method is also resistant to other forms of attack. The method can be used as a black-box watermark (does not require model parameters to verify),  however, the certification bounds only apply to a white-box use case in which the verification can perform inference and test accuracy for a set of trigger images for multiple smoothed versions of the parameters.

Pros -

1. The paper is well written and organized.
2. The paper provides a useful survey of prior art and good motivation for their approach.
3. Unlike prior methods, the method provides a resistance bound for attacks.

Cons -

1. The bound only applies to l2 attacks.
2. The bound only applies to white-box verification.
3. The bound is relatively small.
4. The method reduces accuracy of the trained model.
5. The paper does not provide any direct comparisons to other watermarking methods.
6. The bound is based on empirical estimates which have some uncertainty, so is not actually a true bound.

Cons 1, 3 can be seen as acceptable limitations given this is a step towards certifiable watermarks. Con 4 is par for the course with any watermarking scheme, although the reduction 89.3->86% accuracy for CIFAR-10 is concerning, as that much accuracy loss is a significant deterrent to use of the method and the trend from MNIST CIFAR-10 makes me wonder if larger and more realistic images may show even greater reduction in accuracy. Con 5 is of particular concern for a conference of this tier. If published metrics comparable to the experiments shown in the paper are available, these should be included for side-by-side comparison.

Update: I appreciate the authors response and their hard work in preparing this submission. I also understand that comparisons to prior art are often difficult to obtain. However, I still think further comparisons are warranted to prove out the benefits of this method against other art and whether it can achieve the stated goals for more realistic datasets. The authors did not rebut many of my negative concerns. Most critically, I feel that the method is lacking in a theoretic proof of a strict bound, which is the primary contribution of the paper. For the limitations I mentioned in the review I am leaving my rating score unchanged, but I encourage the authors to continue to develop their approach, which shows promise.

---

> ### Author Response · Authors · 2020-11-21
> **Review Response**
>
> **“The paper does not provide any direct comparisons to other watermarking methods.”**
>
> We did not list any experiments for other watermarking methods because the improvements are either orthogonal to our approaches or not reproducible. More specifically, we did not compare with [2] because the watermark technique does not rely on trigger set recognition and instead directly imprints the watermark onto the model weights. We did not compare with [3] because their approach does not allow for blackbox verification, while our approach does. [5] proposed methods to embed the watermark through the API responses, but that is not applicable when one intends to release the model weights. The only paper that is directly comparable is  [4]. However, it was difficult to reproduce their experiments due to large computational requirements (training hundreds of models is needed) and the fact that they did not release their code. In the end, we chose the initially proposed watermarking method[1], which we believe is a good yardstick for comparison and analysis.
>
> **Concerns that the bound is limited, only applies in white-box verification, and reduces the model's accuracy**
>
> We would like to emphasize the main contribution of the paper is the conceptual framework that it is possible to certify a watermark for neural networks. Similar to the verification and robustness literature, the initial contribution often has quite a number of limitations, but over time, each of the limitations is improved and addressed by the subsequent papers. For example, the concerns about decreased accuracy are reasonable. However, by lengthening the training routine we are able to recover the clean accuracy by more than 1 additional percent. Similar to the adversarial robustness literature, it is likely the case that models exist that have both good certificates and better clean accuracy. Improved training procedures in the future might be more able to find them.
>
> [1] Jialong Zhang, Zhongshu Gu, Jiyong Jang, Hui Wu, Marc Ph Stoecklin, Heqing Huang, and Ian Molloy. "Protecting intellectual property of deep neural networks with watermarking". In Proceedings of the 2018 on Asia Conference on Computer and Communications Security, pp. 159–172, 2018.
> [2] Wang, Tianhao, and Florian Kerschbaum. "Robust and Undetectable White-Box Watermarks for Deep Neural Networks." arXiv preprint arXiv:1910.14268 (2019).
> [3] Li, Huiying, et al. "Piracy Resistant Watermarks for Deep Neural Networks." arXiv preprint arXiv:1910.01226 (2019).
> [4] Lukas, Nils, Yuxuan Zhang, and Florian Kerschbaum. "Deep Neural Network Fingerprinting by Conferrable Adversarial Examples." arXiv preprint arXiv:1912.00888 (2019).
> [5] Szyller, Sebastian, et al. "Dawn: Dynamic adversarial watermarking of neural networks." arXiv preprint arXiv:1906.00830 (2019).

---

### Official Review · AnonReviewer2 · 2020-10-29
**Official Blind Review #2**

**Rating:** 4
**Confidence:** 4

**Review:**

In this paper, the authors propose a certifiable watermarking method for neural networks. The proposed method is based randomized smoothing techniques together with optimizing the model on a dedicated trigger set. The authors show their method can guarantee persistent of watermark examples up to a predefined change in model parameters, in the l2 distance.

Overall, this paper combines several known techniques and is mainly incremental.

As the authors admit, the proposed certification method is somewhat artificial and does not hold in real life scenarios. Additionally, the proposed watermarking embedding method is very similar to [1], [2] but with randomized smoothing technique borrowed from [3].

Questions to the authors:

1) Did you experiment with changing a set of parameters largely while leaving the rest as is? for instance if the model has 1000 parameters, changing 1 of them to be x1000 times larger and leave the rests of the parameters uncharge? that way the l2 change will still be 1, how does that affect model performance?

2) Did the authors experimented with other NN verification/certification methods such as the one proposed in [4]?

3) From a practical point of view, does the model provider need to pre-define \epsilon? If that is the case, how do you suggest to do such thing? empirically?

4) The authors reported results using lr of 0.1, 0.001, and 0.0001. Did the authors also experiment with 0.01?

5) Did the authors try to look for a correlation between watermark removal to accuracy on the in-domain data?

[1] Adi, Yossi, et al. "Turning your weakness into a strength: Watermarking deep neural networks by backdooring." 27th {USENIX} Security Symposium ({USENIX} Security 18). 2018.

[2] Zhang, Jialong, et al. "Protecting intellectual property of deep neural networks with watermarking." Proceedings of the 2018 on Asia Conference on Computer and Communications Security. 2018.

[3] Chiang, Ping-yeh, et al. "Certified defenses for adversarial patches." arXiv preprint arXiv:2003.06693 (2020).

[4] Goldberger, Ben, et al. "Minimal Modifications of Deep Neural Networks using Verification." LPAR. 2020.

---

> ### Author Response · Authors · 2020-11-21
> **Review Response (2/2)**
>
> **"Did the authors try to look for a correlation between watermark removal to accuracy on the in-domain data?"**
>
> Finetuning generally increases the accuracy on in-domain data since the adversary has access to additional data the original owner does not have. For both soft and hard label distillation attacks, the test accuracy in general decreases slightly after the attack as shown in the table below.
>
> ------------------
> Table 2 - Attack method vs test accuracy after attack
>
> **CIFAR10**
>
> Attack Method/Test Accuracy Before Attack/Test Accuracy After Attack
>
> Hard Label (0.001)/**86**/84.16
>
> Hard Label (0.01)/**86**/82.54
>
> Soft Label (0.001)/**86**/84.61
>
> Soft Label (0.01)/**86**/84.63
>
> Fine-Tune (0.001)/86/**88.73**
>
> Fine-Tune (0.0001)/86/**89.36**
>
> **MNIST**
>
> Attack Method/Test Accuracy Before Attack/Test Accuracy After Attack
>
> Hard Label (0.001)/**99.4**/99.15
>
> Hard Label (0.01)/**99.4**/99.23
>
> Soft Label (0.001)/**99.4**/99.3
>
> Soft Label (0.01)/**99.4**/99.27
>
> Fine-Tune (0.001)/99.4/**99.42**
>
> Fine-Tune (0.0001)/99.4/**99.45**

---

> > ### Comment · AnonReviewer2 · 2020-11-24
> > **Updated review**
> >
> > I would like to thank the authors for providing additional experiments and results. I appreciate the authors hard work. After reading the other reviews, the authors response and the updated manuscript, I still find the proposed certification method somewhat artificial and it wont hold under real life scenarios. Since recently there are plenty of methods for watermarking neural networks I expect newly proposed methods to provide more realistic modeling or different perspective to the task.

---

> ### Author Response · Authors · 2020-11-21
> **Review Response (1/2)**
>
> **“Overall, this paper combines several known techniques and is mainly incremental.”**
>
> We would like to emphasize the main contribution of the paper is the conceptual framework that it is possible to certify a watermark for neural networks. Similar to the verification and robustness literature, earlier work often uses simpler techniques, but over time, more sophisticated methods are developed to tailor to the particular application.  Other watermarking methods are purely heuristic and are often broken just by changing the attack  method.  For this reason we think there is merit in studying methods with rigorous security guarantees.
>
> **“the proposed certification method is somewhat artificial and does not hold in real life scenarios.”**
>
> Note that our proposed watermark method is empirically more resilient than previous methods: after watermark removal attack, our blackbox watermark accuracy is higher compared to the baseline watermark by 3-14% on MNIST and 20-50% on CIFAR-10 (see Table 3 in the paper). In the eight scenarios we tested, the baseline watermark only outperforms in two cases where neither the baseline or the proposed method perform well. From this perspective, our watermark holds in real life scenarios in the same way as all of the previous watermarking methods.  However, we acknowledge that the theoretical guarantee that comes with our method is not as strong as its empirical performance.  We hope that the tightness of such guarantees can be improved over time, however as it stands right now this is the only existing theoretical guarantee for certification.
>
> **“Did you experiment with changing a set of parameters largely while leaving the rest as is? for instance if the model has 1000 parameters, changing 1 of them to be x1000 times larger and leave the rests of the parameters uncharge? that way the l2 change will still be 1, how does that affect model performance?”**
>
> Even though our model is certifiably robust with respect to trigger set accuracy, it does not have such a guarantee for the test accuracy since we do not apply randomized smoothing at test time. A large change in a single neuron could decrease the accuracy significantly, or not at all, depending on where the change is made. For example, changes in earlier layers would probably have less impact on prediction compared to changes in later layers.
>
> **“Did the authors experimented with other NN verification/certification methods such as the one proposed in [4]?”**
>
> Thanks for bringing this paper to our attention. The paper is indeed very relevant. However, we did not experiment with the particular approach proposed. To obtain the same l2 certificate for our model using the approach from [4], we would need to solve mixed integer linear programs, which is not currently feasible for the size of the model (Resnet18) that we are considering.  While this approach is not currently tractable for the model sizes used in typical applications, we think this is an interesting direction for future research, and we’ll add a citation and discussion to the paper.
>
> **“From a practical point of view, does the model provider need to pre-define \epsilon? If that is the case, how do you suggest to do such a thing? Empirically?”**
>
> In general, a larger epsilon is always better. However, training the model to be robust to an infinite epsilon is likely impossible. In practice, one can look at the empirical norm change during training to appropriately select epsilon. For example, if one wants the watermark to be robust against 1 epoch of finetuning, then one could select the approximate l2 norm change after 1 epoch of removal attack to be the epsilon. However, as the epsilon increases, the training would likely become more difficult and the generalization would also decrease, both of which the owner has to take into account when making the epsilon selection.
>
> **“The authors reported results using lr of 0.1, 0.001, and 0.0001. Did the authors also experiment with 0.01?”**
>
> Different attack methods require different learning rates to be successful. We selected a minimal learning rate for each attack method such that the attack is able to succeed for the baseline method. However, for completeness, we ran an attack with the 0.01 attack as suggested, and found our method still consistently outperforms the baseline blackbox watermark.
>
> --------------------------
> Table 1 - Watermark retention with new attack settings
>
> Dataset/Attack Method/Baseline Black Box Watermark Accuracy/Our Black Box Watermark Accuracy
>
> CIFAR 10/Hard Label (0.01)/9.38/**51.56**
>
> CIFAR 10/Soft Label (0.01)/50.00/**81.25**
>
> MNIST/Hard Label (0.01)/46.88/**59.38**
>
> MNIST/Soft Label (0.01)/98.44/**100.00**
>
> --------------------------

---

### Official Review · AnonReviewer3 · 2020-10-30
**Certification provided by this method should be clearly stated.**

**Rating:** 6
**Confidence:** 4

**Review:**

The proposed method exploits the randomized smoothing techniques for a certified watermark of neural networks. The idea itself is novel and interesting. To the best of the reviewer's knowledge, no one has ever used randomized smoothing for neural network watermark. Different from the defense against adversarial example, in the case of watermark detection, not only the detection accuracy but false detection of non-watermarked models should be considered. If my understanding is correct, the proposed method does not give certification on the false detection.  Since the proposed method is quite close to adversarial training, one concern is that models trained with adversarial training might be falsely detected as the watermarked model.

 Since the subject of this study is certified watermarks, its certification should be clearly stated in the form of Theorem. If my understanding is correct, Col. 1 simply certifies that the lower bound on the trigger set accuracy. Then, for the l2-constrained adversary, what is certified for the model trained with Alg. 2?

Suppose we can certify that the trigger set accuracy does not drop below 51% as long as parameters do not move more than an l2 distance of 1. Let’s say, given a suspicious model, the trigger set accuracy was 55%. Then, what can we say?  Can we say that the suspicious model is truly the watermarked model?  Can we say that models without watermark cannot attain this trigger set accuracy?

Minor:
Is no condition on f required to have  Corollary 1? Does Corollary 1 work with any f?

---

> ### Author Response · Authors · 2020-11-21
> **Clarification on the watermark recognition threshold**
>
> **“one concern is that models trained with adversarial training might be falsely detected as the watermarked model.”**
>
> There are two major differences between an adversarially trained model and a model trained with our approach. First, a network trained with adversarial training tries to classify regular examples correctly while our network tries to classify trigger set examples correctly. The regular examples and trigger set examples have very different signals and are unlikely to result in similar models. Second, our approach applies perturbation in the parameter space, while adversarial training applies perturbation in the input space. These two differences make false detection of adversarially trained models very unlikely.
> Empirically, we tested our trigger set examples on an adversarially trained model, and found that adversarial training does not yield any benefits to trigger set classification.
>
> **“If my understanding is correct, Col. 1 simply certifies that the lower bound on the trigger set accuracy. Then, for the l2-constrained adversary, what is certified for the model trained with Alg. 2?”**
>
> For the l2-constrained adversary,  we can certify that it is not possible for the adversary to decrease the  trigger set accuracy below the watermark recognition threshold. Note that we set the watermark recognition threshold ourselves.
>
> **“Let’s say, given a suspicious model, the trigger set accuracy was 55%. Then, what can we say? Can we say that the suspicious model is truly the watermarked model? Can we say that models without watermark cannot attain this trigger set accuracy?”**
>
> We think this is a deeper question of what makes the watermark recognition threshold valid, for any watermarking technique. In our threat model, we assume that the adversary does not have access to our arbitrarily chosen trigger set pattern, so a model trained without access to these trigger set labels is extremely unlikely to correctly predict trigger set labels. If this assumption holds, then the probability of getting any materially significant proportion of trigger set examples correct would be incredibly small. For example, if we have 100 trigger set examples, the likelihood of getting more than 25 correct on CIFAR-10 would be smaller than 2e-5 if based on random chance.
>
> **“Minor: Is no condition on f required to have Corollary 1? Does Corollary 1 work with any f?”**
>
> Corollary 1 works for any measurable function f.   We will clarify this in the paper.

---

### Decision · Program_Chairs · 2021-01-07
**Final Decision**

**Decision:**

Reject

**Comment:**

While it’s commonly acknowledged that the paper is well written, the reviews are a bit split: R3 and R1 are mildly positive/negative, respectively, R2 and R4 both voted for reject. R2 asked many questions regarding experiments, which were addressed in the details in the rebuttal. R4 raised 6 questions regarding the bound, and the authors only answered some of them in the rebuttal. R4 felt “the method is lacking in a theoretic proof of a strict bound, which is the primary contribution of the paper”. Both R1 and R4 pointed out the proposed algorithm is not practical as expected, especially the results on larger scale such as ImageNet are missing.

The AC cannot agree with the authors’ argument that the contribution of the paper is “a conceptual framework that it is possible to certify a watermark for neural networks” in responding to such criticisms. It’s indeed very important for this conceptual framework to be proven valuable through thorough experiments and solid comparisons.